# Spatial distribution characteristics and influencing factors of China martial arts schools based on Baidu map API

**Pengfei Yu**[1], **Xiaoming Yang**[2]*, **Qi Guo**[3], **Jianliang Guan**[4], **Guohua Chen**[5]

**1** Department of Physical Education, School of Physical Education, East China University of Technology, Nanchang, Jiangxi, China, **2** College of Physical Education, Putian University, Putian, Fujian, China, **3** Department of Sports Studies, Faculty of Educational Studies, Universiti Putra Malaysia, Serdang, Selangor, Malaysia, **4** Department of Economics and Management, Shanghai University of Sport, Shanghai, China, **5** Department of Physical Education, School of Physical Education, East China University of Technology, Nanchang, Jiangxi, China,

* haloryu5@gmail.com

**Data availability statement:** All relevant data are within the paper and its Supporting

## Abstract

This paper examines the spatial distribution pattern and influencing factors of Martial Arts Schools (MASs) based on Baidu map data and Geographic Information System (GIS) in China. Using python to obtain the latitude and longitude data of the MASs through Baidu Map API, and with the help of ArcGIS (10.7) to coordinate information presented on the map of China. By harnessing the geographic latitude and longitude data for 492 MASs across 31 Provinces in China mainland as of May 2024, this study employs a suite of analytical tools including nearest neighbor analysis, kernel density estimation, the disequilibrium index, spatial autocorrelation, and geographically weighted regression analysis within the ArcGIS environment, to graphically delineate the spatial distribution nuances of MASs. The investigation draws upon variables such as martial arts boxings, Wushu hometowns, intangible cultural heritage boxings of Wushu, population education level, Per capita disposable income, and population density to elucidate the spatial distribution idiosyncrasies of MASs. (1) The spatial analytical endeavor unveiled a Moran's I value of 0.172, accompanied by a Z-score of 1.75 and a P-value of 0.079, signifying an uneven and clustered distribution pattern predominantly concentrated in provinces such as Shandong, Henan, Hebei, Hunan, and Sichuan. (2) The delineation of MASs exhibited a prominent high-density core centered around Shandong, flanked by secondary high-density clusters with Hunan and Sichuan at their heart. (3) Amongst the array of variables dissected to explain the spatial distribution traits, the explicative potency of 'martial arts boxings', 'Wushu hometowns', 'intangible cultural heritage boxings of Wushu', 'population education level', 'Per capita disposable income', and 'population density' exhibited a descending trajectory, whilst 'educational level of the populace' inversely correlated with the geographical dispersion of MASs. (4) The entrenched regional cultural ethos significantly impacts the spatial layout of martial arts institutions, endowing them with distinct regional characteristics.

information files. The Baidu Map API data URL is from (https://api.map.baidu.com/lbsapi/getpoint/).

**Funding:** This study was funded by the Research on the Optimization Path of University Physical Education Curriculum System Driven by New Quality Productivity (Number: DHJG-24-71) fund project.

**Competing interests:** The authors have declared that no competing interests exist.

## 1. Introduction

Chinese Martial Arts (CMAs), also known as 'Kung Fu' and 'Wushu,' transcend the mere transmission of physical skills, embodying a rich tapestry of cultural heritage, philosophy, and psychological well-being [1–4]. CMAs have a deeply ingrained presence in Chinese society, supported by numerous clubs and schools dedicated to the training of martial arts athletes and students [1]. While 'Martial Arts' in China is frequently regarded as a cultural symbol, activities such as Western boxing and wrestling are typically classified as 'sport' or 'combat sports' [5]. The origins of traditional Wushu culture can be traced back to the Shang Dynasty (1600–1046 BC) [1]. Notably, 43 percent of overseas respondents believe that Chinese martial arts are among the elements that best represent Chinese culture [6]. Martial Arts Schools (MASs) serve as crucial institutions for the preservation and dissemination of Wushu culture, acting as the primary platform for the development of basic martial arts education in China [7]. These schools not only play a pivotal role in safeguarding and promoting China's illustrious traditional culture but also hold an important position within the framework of the compulsory educational system [8–11]. Ideally, martial arts education integrates Chinese classical philosophy with martial arts skills [12]. However, in recent years, martial arts have encountered significant challenges and deficiencies within China's compulsory education system. The marginalization of martial arts within the educational framework remains a major issue [2,13], and martial arts cultural education lags far behind other areas of 'arts' education. Despite the breadth of research on martial arts education, most scholars have concentrated on the timeline of martial arts development and the impact of educational policy documents on martial arts school education [2,11,13,14].Since the General Administration of Sport of China conducted a nationwide investigation into martial arts education in primary and secondary schools in 2009, there has been a scarcity of empirical studies examining the overall state of martial arts education in China. The results of the 2009 investigation revealed that the situation of martial arts education in schools across the country was not optimistic, and martial arts schools were excluded from the study[8,15]. Over the past two decades, there has been little research on the development of these schools receiving minimal scholarly attention. Currently, the field of martial arts education lacks comprehensive data statistics and resource allocation guidelines. As a result, relevant departments face difficulties in providing effective solutions and policy support for the challenges encountered by MASs [11].

Past research on spatial distribution characteristics within China has yielded significant advancements across various fields. Scholars have primarily focused on exploring the intricate spatial interrelationships among variables such as per capita GDP, environmental pollution control, and per capita energy production across different provinces and cities [16–19]. Additionally, studies have examined the spatial distribution characteristics of Wushu hometowns and tourism-centric towns by integrating factors, including the natural environment, topography, river systems, climate, cultural influences, the Hu Line, and transportation networks [16,20–24]. Population resources, economy, and the natural environment have also been extensively utilized as primary variables to assess spatial distribution, significantly impacting the spatial distribution of tourist towns and ski resorts in China [21,25,26]. This analysis further encompasses the temporal-spatial evolution and influencing factors of coordinated development within the Population Resources, Economy, and Environment system in China [16]. Additionally, some scholars have employed Baidu migration data to analyze the spatial network of Chinese population mobility [27,28]. Collectively, these foundational studies on martial arts education and spatial distribution in China provide a robust basis for further exploration in this field.

Previous studies about Baidu Data, many scholars have separately utilized Baidu indices, Baidu migration data, and Baidu Map Point of Interest data to analyze various phenomena, such as the

relationship between PM2.5 exposure and anxiety at the prefectural level in China [17], the network patterns of the Zhongyuan Urban Agglomeration in China [28], and the spatial distribution characteristics of agritourism in Henan Province [29]. These studies offer valuable insights for this research, which utilizes Baidu Map API data to collect the names and geographic coordinates of martial arts schools in China. Moreover, the use of Baidu Map API data effectively addresses the technical challenges associated to the difficulty of map data collection in this study.

## 2. Data and methods

### 2.1. Data sources

**2.1.1. Selection of regions and sources of MASs geographical information.** We chose "martial arts school" and "Wushu school" as the search terms; Leveraging the Python programming language, an extensive dataset encompassing the geographical metadata of 1,986 MASs was meticulously harvested from Baidu Maps (https://api.map.baidu.com/lbsapi/getpoint/). We distilled this dataset to the precise names, along with the geographical coordinates—latitude and longitude-of 492 MASs, as cataloged by the end of May 2024. The latitude and longitude coordinates of the martial arts schools, after screening, can clearly reflect the number of martial arts schools at the provincial scale. It is important to note that this study deliberately excludes the regions of the Hong Kong Special Administrative Region, the Macao Special Administrative Region, and Taiwan Province from its geographical scope. The economic and educational systems in these regions differ significantly from those in mainland China, making it inappropriate to analyze the factors influencing the spatial distribution of martial arts schools on the same basis. Therefore, these regions were not included in the analysis.

**2.1.2. Data screening.** To screen out the necessary information of MASs' name, we used keywords to carry out three times of data elimination: (1) The first deletion keywords: 'Committee, dance school, club, dormitory, Admission Office, number building, South gate, North Gate, East Gate, West Gate, residential area, shopping plaza, hotel, eloquence school, sports training school, martial arts hall, training center, association, Guild Hall, education center'. (2) The second deletion keywords: 'Company, Center, Memorial, Media, Training base, National Academy, Dragon and Lion Club, Workshop, studio, Martial arts group, Martial arts, Taekwondo, Sanda gym, Tutorial class, General Association'. (3) The third cleaning keywords: 'Company, center, memorial hall, wake teacher, supplies, media, training base, National Academy, Dragon and Lion Club, workshop, martial arts group, martial arts, Taekwondo, Sanda gym, tutorial class, General Association'.

### 2.2. Study methods

In this study, the methods, namely, Nearest Neighbor Index, Geographical Concentration Index, Disequilibrium Index, Geographical Weighted Regression, Nuclear Density Analysis, Moran's I and Getis-Ord Gi*. Were employed to analyze the traits of the participants and to elucidate their spatial aggregation under a significance level of $P < 0.05$. The resulting data were visualized using the following equations.

**2.2.1. Nearest neighbor index (NNI)** [30].

$$R = \frac{r_1}{r_E} \tag{1}$$

$$r_E = \frac{1}{2}\sqrt{n/A} \tag{2}$$

In formula (1), $R$ stands for nearest neighbor index; $r_1$ represents the nearest proximity; $r_E$ stands for theoretical nearest neighbor distance. In formula (2), $n$ represents the number of MASs; $A$ represents the size of the study area. When $R = 1$, it indicates that Chinese MASs belong to the random distribution type. When $R > 1$, it belongs to the uniform distribution type. When $R < 1$, it belongs to the cluster distribution type.

### 2.2.2. Geographical concentration index (GCI) [31,32].

$$G = 100 \times \sqrt{\sum_n^{i=1} \left(\frac{P_i}{Q}\right)^2} \tag{3}$$

$G$ represent the geographical concentration index, $P_i$ indicates the number of MASs in the $i$ province (city or autonomous regions), the $n$ Indicates the number of provinces (city or autonomous regions), the $Q$ Represents the total number of MASs. The value of $G$ is (0,100). The larger the $G$ value is, the more concentrated the distribution of Wushu school is. The smaller the $G$ value is, the distribution of MASs tends to be dispersed.

### 2.2.3. Nuclear density analysis (NDA) [21].

$$f_h(x) = \frac{1}{nh} \sum_{i=1}^{n} (\frac{x - x_i}{h}) \tag{4}$$

$f_h(x)$ represents the kernel density function, $n$ is the number of MAS, $x - x_i$ is the distance from $x$ to $x_i$, $h$ represents the width, $h > 0$. The greater the value of $f_h(x)$, the denser the spatial distribution of MASs.

### 2.2.4. Disequilibrium index (DI) [32].

$$S = \frac{\sum_{i=1}^{n} Y_i - 50(n+1)}{100 \times n - 50(n+1)} \tag{5}$$

$n$ indicates the number of provinces (city or autonomous regions), $Y_i$ is the accumulated percentage of the $i$th digit, The value of $S$ is between 0 and 1, and the increase of the value indicates that the spatial distribution of MASs is more unbalanced. When $S = 0$, it means that MASs are evenly distributed in each province, and when $S = 1$, it means that MASs are all clustered in a certain province (city or autonomous regions)

### 2.2.5. Local spatial autocorrelation (LSA) [25,33].

$$G(d) = \frac{\sum_i \sum_j w_{ij}(d) x_i x_j}{\sum_i \sum_j x_i x_j} (i \neq j) \tag{6}$$

$x_i x_j$ is the observed value of $i$ and $j$ in the region respectively; $w_{ij}$ is the spatial weight matrix. When $G(d)$ is positive and significant, it indicates that the values around $i$ are relatively high and belong to high-value spatial agglomeration. When $G(d)$ is negative and significant, it indicates that the values around $i$ are relatively low and belong to low-value spatial agglomeration. When Moran's I > 0, it means that the unit observations are spatially clustered. When Moran's I < 0, the observed values show discrete distribution. When Moran's I = 0, it indicates that the observed values are randomly distributed.

### 2.2.6. Geographical weighted regression (GWR) [25].

$$y_i = \beta_0(u_i, v_i) + \sum_{k=1}^{n} \beta_k(u_i, v_i) x_{ik} + \varepsilon_i \tag{7}$$

$(u_i, v_i)$ represents the geographical coordinates of the $i$ th sample space unit, $\beta_k(u_i, v_i)$ represents the $k$ th regression coefficient of sample space unit $i$, $y_i$ represents the number of martial arts schools in unit $i$ of the sample space, $x_{ik}$ represents the observed value of the $k$ th influencing factor in the spatial unit $(u_i, v_i)$, In this study, $k$ represents factors that may affect the spatial distribution of MASs, such as the number of Wushu boxings in each province, the distribution of population, the education level of population and the number of Wushu hometowns, $n$ represents the number of influencing factor, $\varepsilon_i$ is the residual term, which follows the normal distribution, and the smaller the residual, the hig.er the fit of the regression equation.

## 3. The spatial distribution results analysis of MASs

### 3.1. Spatial agglomeration type

The spatial distribution analysis of (MASs) reveals a Z-score of 1.75390, indicating that the probability of this spatial distribution occurring randomly is less than 10%. This finding, as illustrated in Fig 1, confirms that the spatial distribution of MASs is significantly clustered (Fig 1). Moreover, the results at the provincial level demonstrate distinct variations in the distribution of MASs across different regions. These insights underscore the critical importance of investigating the spatial distribution characteristics of MASs, as such research holds substantial value for understanding regional disparities and informing educational policy. The statistics and picture show that (Table 1 and Fig 2), 81.09% of the country's martial arts schools are distributed in North China, East China, Central China, and Southwest China. These regions not only have a long history of social development, but also are the main gathering places for the development and inheritance of martial arts skills and culture, with the largest number of martial arts hometowns, martial arts boxings, and martial arts training groups in the country. The number of martial arts schools in Shandong, Henan, Hebei, and Shanxi provinces alone has reached 199, accounting for 40.44% of the total number of MASs in China. Secondly, the number of MASs in Hunan, Sichuan and Guizhou provinces in southern Central China and northern Southwest China is 84, accounting for 17.07% of the national total. The number of MASs in Southern China, Northeastern China and Northwestern, and Northwestern China is 80, accounting for 5.28%, 7.31% and 3.65% of the country, respectively. The nearest neighbor index is used to analyze the spatial distribution types of MASs. It is found that the average observation distance of CMASs is 28.16 km, the theoretical nearest neighbor distance is 70.36 km, and the nearest neighbor index $R = 0.40$, which indicates that the spatial distribution of CMASs belongs to the cluster distribution type.

### 3.2. The disequilibrium index analysis

By calculating the imbalance index S = 0.55 (<1) of MASs at the provincial scale, it can be found that the MASs in each province present an unbalanced distribution. In combination with the Lorentz curve (Fig 3) of MASs in various provinces in China, the combination of MASs in Hebei, Shandong, and Henan province accounts for 38.21% of the total number of MASs in China, which further confirms the unbalanced spatial distribution of Chinese MASs (Table 1). This unbalanced distribution reflects that the country's martial arts cultural resources are mainly concentrated in the Shandong, Henan, and Hebei regions. As the political, economic, and cultural centers for thousands of years, these regions have profound martial arts cultural deposits, rich martial arts boxing types and numerous martial arts talents, which provide incomparable innate advantages for the development of MASs. Secondly, the education departments and sports departments in Shandong, Henan, and

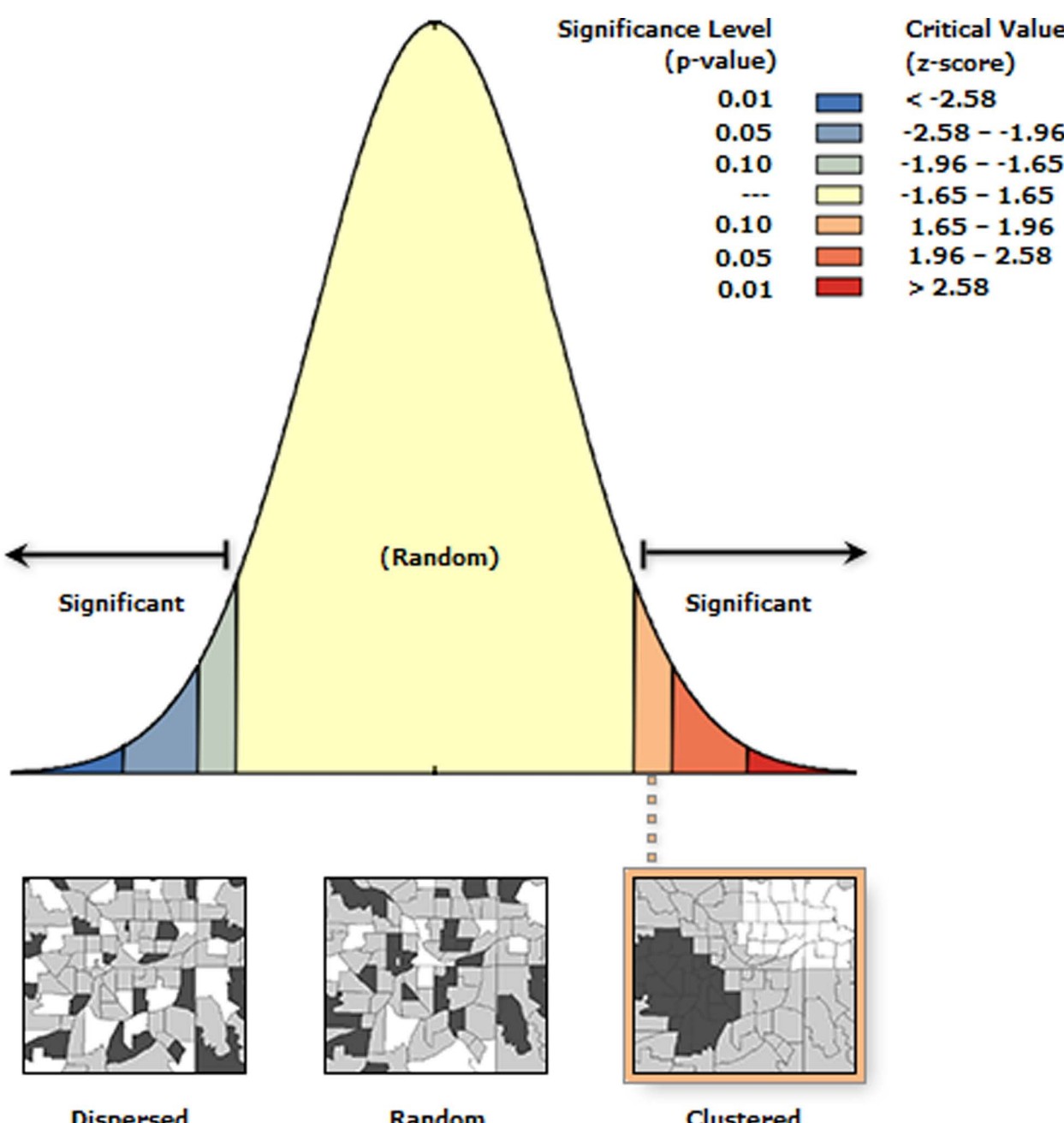

**Fig 1. The MASs spatial distribution type diagram.** (A) The evaluation of spatial distribution distance of martial arts schools was derived from the Moran's I calculation tool of ArcGIS software. (B) The P-value was significant at the 10% level, indicating that Wushu schools presented agglomeration distribution characteristics in local space, and a small part presented random distribution characteristics.

Hebei regions have given strong support to the development of MASs, and these schools have received a lot of policy benefits and support in terms of funds, policies, and student conditions. Finally, from the perspective of 'martial arts culture circle' [5], people in Shandong, Henan, and Hebei have a deep understanding of martial arts culture after long-term immersion in martial arts culture, so they have a strong subjective will and responsibility to practice and inherit martial arts [3].

**Table 1. Statistics on the number of MASs in China's provinces.**

| Province | Quantity | Percentage | Cumulative percentage | Province | Quantity | Percentage | Cumulative percentage |
|---|---|---|---|---|---|---|---|
| Shandong | 73 | 14.84 | 14.84 | Anhui | 10 | 2.03 | 86.99 |
| Henan | 70 | 14.23 | 29.07 | Jiangxi | 9 | 1.83 | 88.82 |
| Hebei | 45 | 9.15 | 38.21 | Fujian | 9 | 1.83 | 90.65 |
| Hunan | 42 | 8.54 | 46.75 | Jilin | 7 | 1.42 | 92.07 |
| Sichuan | 32 | 6.50 | 53.25 | Guangxi | 7 | 1.42 | 93.50 |
| Jiangsu | 28 | 5.69 | 58.94 | Beijing | 7 | 1.42 | 94.91 |
| Zhejiang | 18 | 3.66 | 62.60 | Shanghai | 6 | 1.22 | 96.14 |
| Heilongjiang | 16 | 3.25 | 65.85 | Shanxi | 6 | 1.22 | 97.35 |
| Guangdong | 16 | 2.64 | 69.11 | Inner Mongolia | 6 | 1.22 | 98.58 |
| Liaoning | 13 | 2.24 | 71.75 | Hainan | 3 | 0.61 | 99.19 |
| Chongqing | 11 | 2.24 | 73.98 | Tibet | 2 | 0.41 | 99.59 |
| Yunnan | 11 | 2.24 | 76.22 | Tianjin | 1 | 0.20 | 99.80 |
| Shanxi | 11 | 2.24 | 78.45 | Ningxia | 1 | 0.20 | 100.00 |
| Hubei | 11 | 2.24 | 80.69 | Xinjiang | 0 | 0.00 | 100.00 |
| Gansu | 11 | 2.24 | 82.92 | Qinghai | 0 | 0.00 | 100.00 |
| Guizhou | 10 | 2.03 | 84.96 | —— | —— | —— | —— |

## 3.3. The spatial distribution density analysis

We use ArcGIS10.8 to conduct a nuclear density analysis of MASs in China, and the results showed the spatial agglomeration pattern of MASs and formed one high-density and two sub-high-level density agglomeration areas (Fig 4). (1) In the high-density areas, the southwest of Shandong and the north of Henan are the core areas, and their influence radiates to Anhui and Jiangsu. At the same time, Shandong and Hebei interact with each other, and its influence radiates to Tianjin and Beijing. From the number of martial arts schools in Beijing, Tianjin, Jiangsu, Shanxi and other regions and their nuclear density values, the high-density MASs clustered in Shandong, Henan, and Hebei regions have obvious driving effect on MASs in neighboring provinces and municipalities. (2) Among the two sub-high-density regions, the central China sub-high-density region with Hunan as the core radiates southwest to Guizhou, Chongqing, and other regions. At the same time, the southwest of China and Yunnan regions with Sichuan as the core also extend to Guizhou in the east and Chongqing in the north, showing a corresponding trend. Due to the influence of terrain environment and national culture in Yunnan, Guizhou, and Sichuan regions [5,34], MASs have been affected to a certain extent in terms of radiation range and breadth. According to the survey, Sichuan, and Hunan are the gathering places of 'Emei Wushu' and 'Wudang Wushu' respectively, with a high degree of boxings richness and a relatively dense distribution of MASs. (3) The spatial distribution of MASs in the country generally presents a trend of more east and less west, dense in the north and sparse in the south, which is closely related to the Hu's line (the dividing line of China's population development level and economic and social pattern).

## 3.4. Spatial autocorrelation hot and cold spot analysis

**3.4.1. Moran's I value analysis of MAS spatial autocorrelation test.** It ranges between −1 to 1 of Moran's I value, usually interpreted as a spatial autocorrelation coefficient. At a given level of significance, through Z test, if Moran's I value is significantly positive, there is positive spatial autocorrelation, indicating the value of the same property together; if

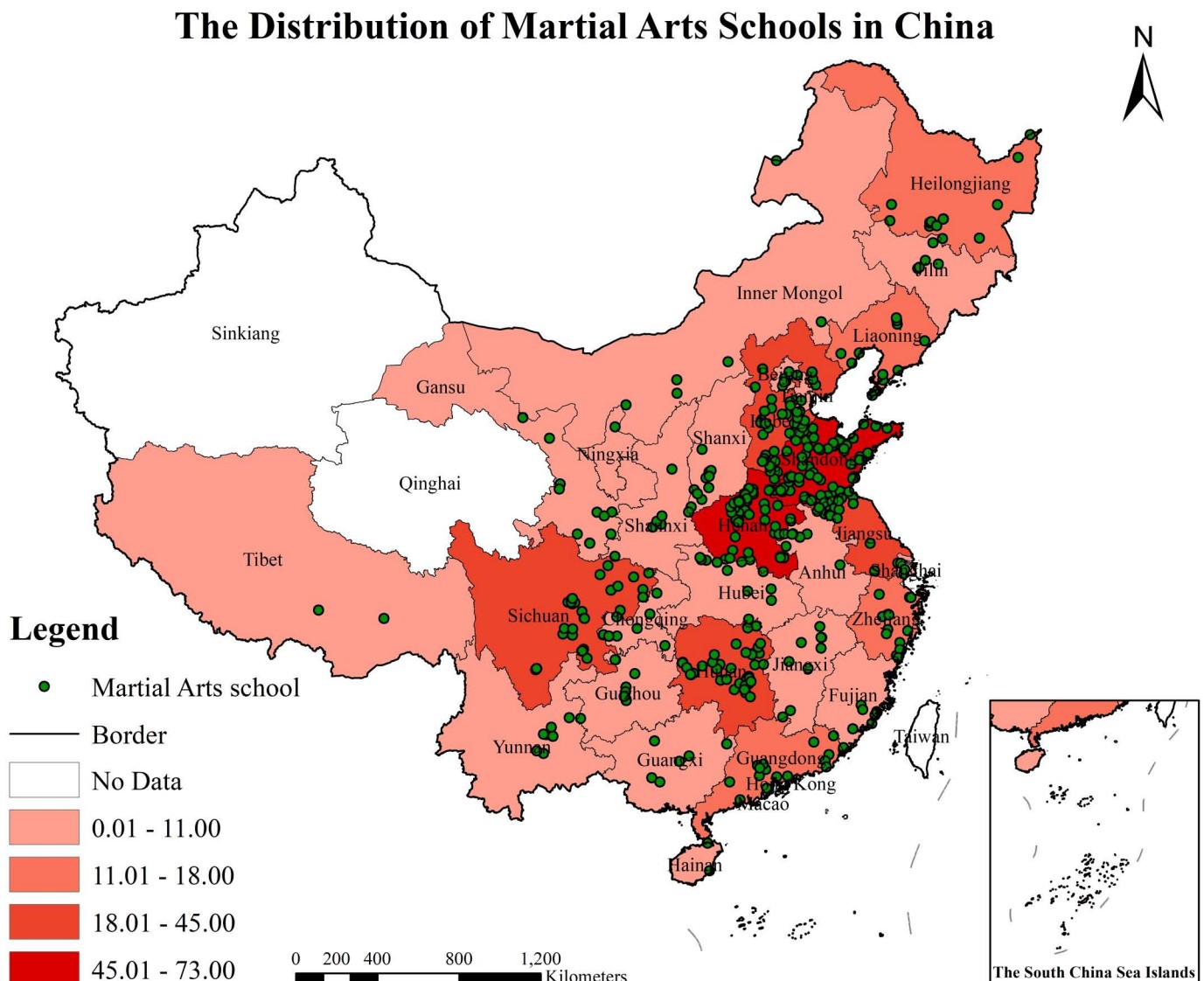

**Fig 2. The distribution of martial arts schools in China** ( http://bzdt.ch.mnr.gov.cn/browse.html?picId=%224o28b0625501ad13015501ad2bfc0272%22). (A) The map is based on the standard map service website of the National Bureau of Surveying, Mapping and Geographic Information Service No.GS (2019)1682 standard map drawing, the base map has not been modified. (B) Excluding data on martial arts schools in Hong Kong, Macau, and Taiwan.

significant Moran's I value is negative, there is negative spatial autocorrelation, indicating the value of different properties together. The Moran's I = 0.172, Z-score = 1.754, and P value = 0.079 (Table 2), indicating that there is positive spatial autocorrelation the data distribution and the clustered model. Based on the data, this study further divides its spatial distribution into Hot Spots (99% Confidence), sub-Hot Spot (95% Confidence), low-Hot Spot (90% Confidence), Not Significant areas, Cold Spot (99% Confidence), sub-Cold Spot (95% Confidence), and low- Cold Spot (90% Confidence) (Fig 5). Among the martial arts schools in China, Shandong Province is a hot area with 99% confidence; the sub-hot spots are in Henan, Hebei, and Anhui province, with 95% confidence. From the perspective of hot and cold points, although Hunan, Sichuan and Jiangsu regions show

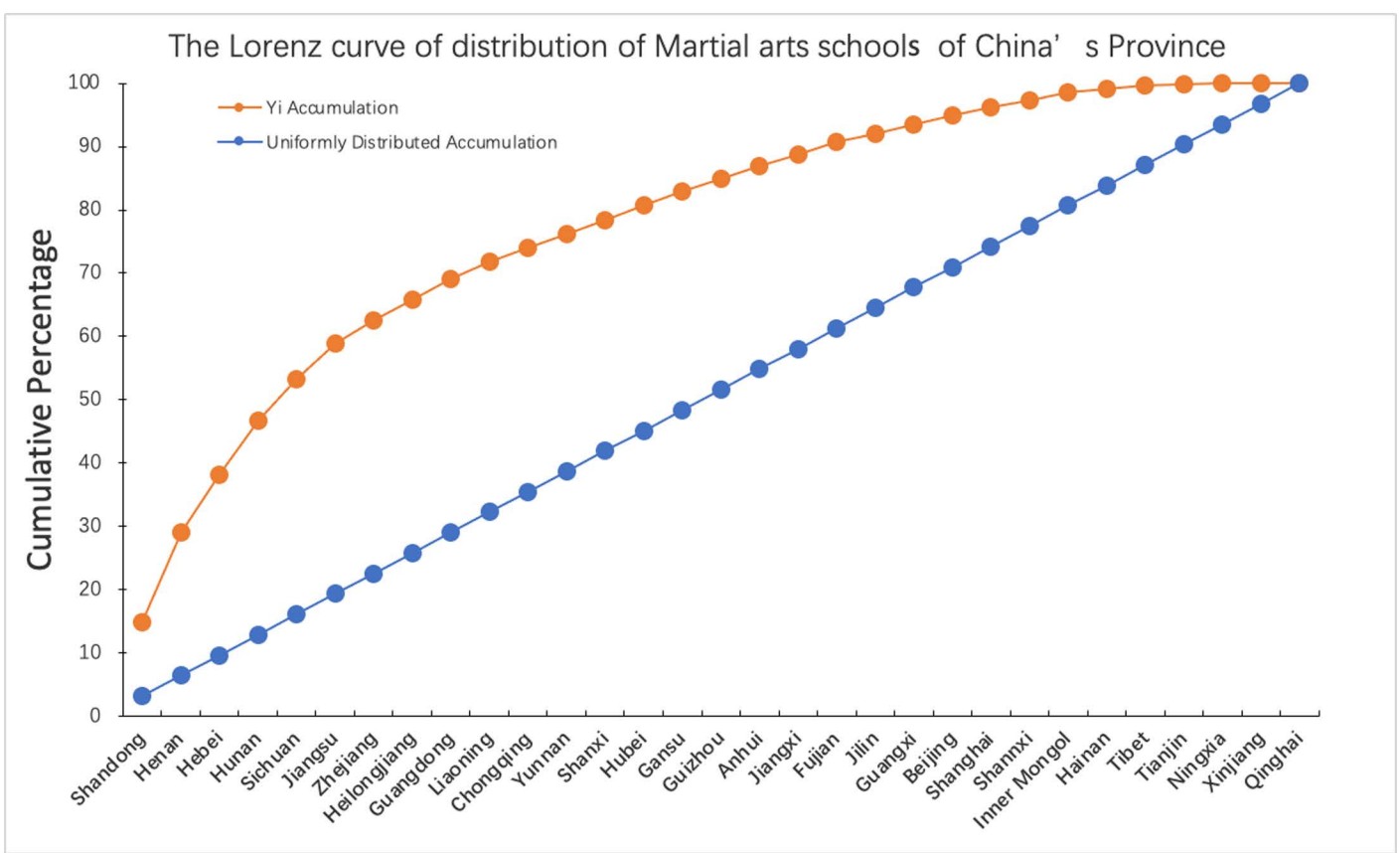

**Fig 3. The Lorenz curve of distribution of Wushu schools of China's Province.** (A) The data for the Lorenz curve comes from the cumulative sum of the proportion of martial arts schools in 31 provinces of China. (B) Excluding data on martial arts schools in Hong Kong, Macau, and Taiwan.

an insignificant trend, it can only indicate that their confidence intervals are not up to 90%. Combined with the point density of martial arts schools, the distribution of MASs in the above provinces lies in the low hot and non-significant range. The calculation results show that except for the absence of MASs' data in Xinjiang, Qinghai, Hong Kong, Macao and Taiwan, the data of MASs in most regions show no significant characteristics between the cold and hot spots. The comparison results show that the martial arts schools with the highest popularity and the richest quality resources in the country are mainly concentrated in Shandong, Henan, Hebei, and Anhui, which are also the main parts of hot spots and sub-hot spots.

## 4. The selection of influencing factors and the analysis of geographical weighted regression results

### 4.1. The influencing factors

The spatial clustering and distribution characteristics of MASs are the result of multiple factors. Therefore, according to the spatial distribution characteristics of MASs, we consider regional characteristics, historical culture, and population distribution as the influencing factors. We further take intangible cultural heritage boxings of Wushu, population education level, and population density, Per capita disposable income, and martial arts boxings quantity as the independent variables of the possible influencing factors,

# Kernel Density Analysis of Martial Arts Schools in China

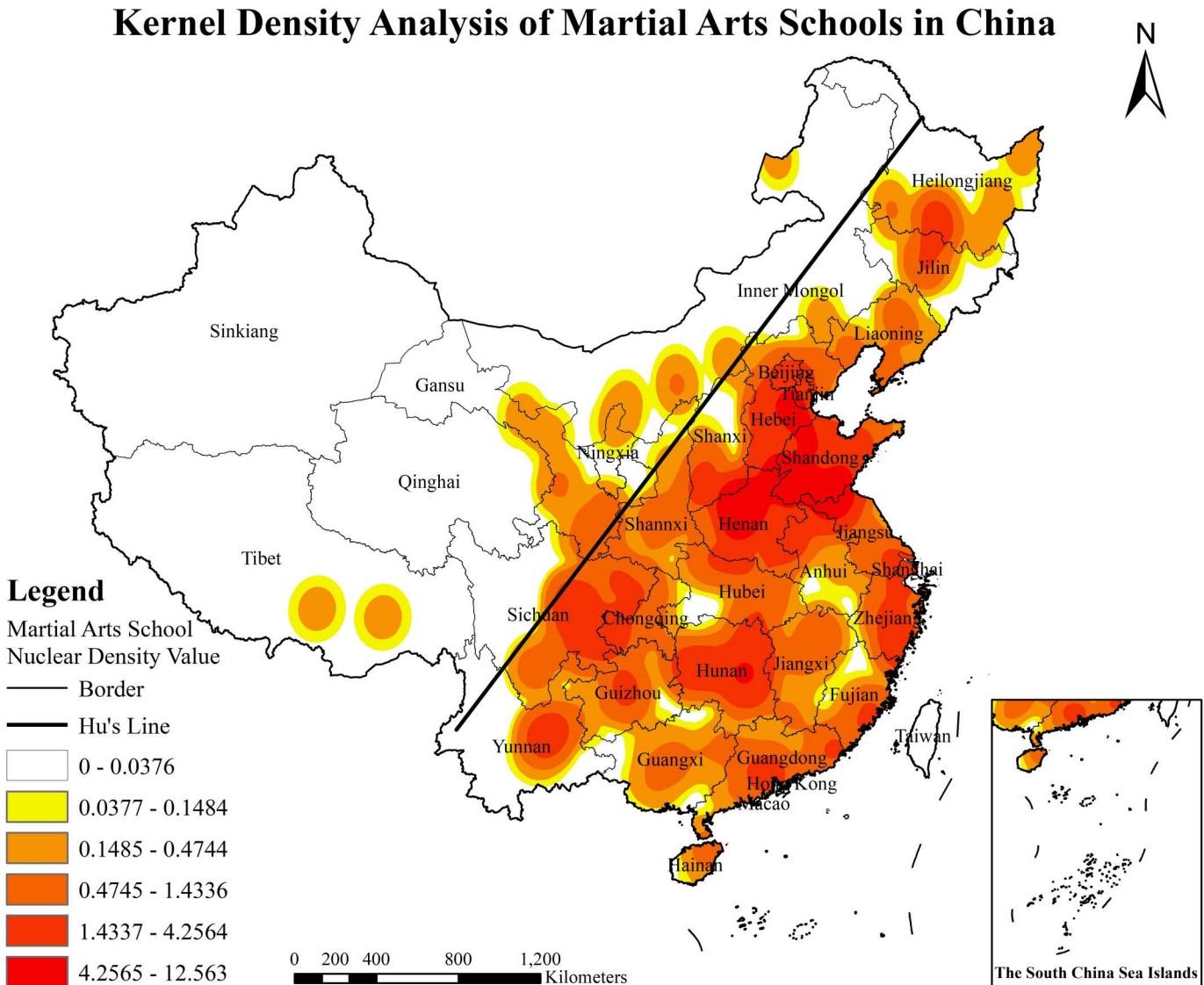

**Fig 4.  The kernel density analysis of arts schools in China** (http://bzdt.ch.mnr.gov.cn/browse.html?picId=%224o28b0625501ad13015501ad2bfc0272%22). (A) The map is based on the standard map service website of the National Bureau of Surveying, Mapping and Geographic Information Service No.GS (2019)1682 standard map drawing, the base map has not been modified. (B) The map projection for the martial arts school was obtained from the Baidu Map API using latitude and longitude data in Python. (C) Using ArcGIS software and applying the kernel density calculation formula (4), the spatial density projection of martial arts schools is derived.

**Table 2.  Moran's I index and its test results of martial arts school.**

| Name | Moran's I | Z-score | P-value |
|---|---|---|---|
| Parameter values | 0.172379 | 1.753902 | 0.079447 |

and the number of MASs in each province was taken as dependent variable to construct a GWR model [8,25,27,31,35–37]. Analyze the different influence of various factors on the spatial distribution of MASs. The meaning of variables in regression model and its algorithm (Table 3).

# The Hot and Cold Point Distribution of Martial Arts Schools in China

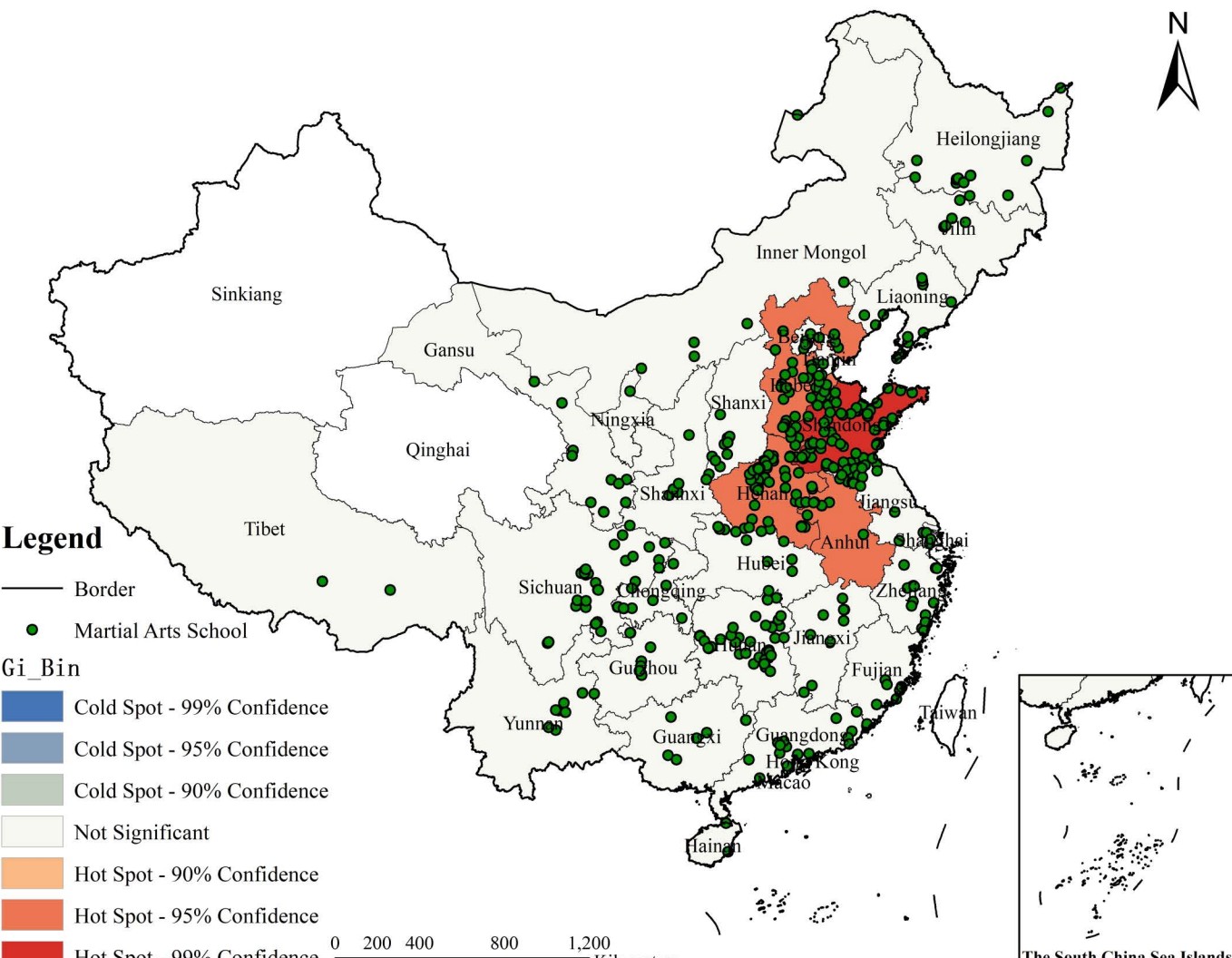

**Fig 5. The hot and cold point distribution of martial arts schools in China** (http://bzdt.ch.mnr.gov.cn/browse.html?picId=%224o28b0625501a-d13015501ad2bfc0272%22). (A) The map is based on the standard map service website of the National Bureau of Surveying, Mapping and Geographic Information Service No.GS (2019)1682 standard map drawing, the base map has not been modified. (B) The map projection for the martial arts school was obtained from the Baidu Map API using latitude and longitude data in Python. (C) Using ArcGIS software and applying the kernel density calculation formula (6), the spatial autocorrelation characteristics of martial arts schools are derived, and further classified into hot spot and cold spot areas.

## 4.2. The ordinary least squares model result analysis

It was found that VIF of each factor was less than 10, indicating that there was no collinearity between the factors. The P-value test is significant at 10% level, indicating that the model is robust. The test results of the five factors of the number of provincial Wushu intangible cultural heritage of boxings, the number of Wushu boxings, the population density, the education level of the population, and per capita disposable income of each province are significant and have different degrees of influence on the number and spatial distribution of MASs. The coefficient calculated by OLS model shows that the education level of the population is hardly correlated with the number of MASs, the number of provincial Wushu

**Table 3. The meaning and calculation method of variables in GWR model.**

| Type | Index | Computing method |
|---|---|---|
| Dependent variable | Number of MASs | Average number of MASs in each province |
| Dependent variable | Number of Wushu intangible cultural heritage boxings of each province | Intangible Wushu boxings number |
| Dependent variable | Population education level | Population average educational level of each province |
| Dependent variable | Population density in each province | Average population density of each province |
| Dependent variable | Per capita disposable income of each province | Per capita disposable income of each province |
| Dependent variable | Martial arts boxings number of province | Average number of martial arts boxings in each province |

intangible cultural heritage of boxings, the number of martial arts boxings are positively correlated, and Per capita disposable income of each province is negatively correlated with the number of MASs. From the numerical value, the number of martial arts boxings, and the number of provincial Wushu intangible cultural heritage of boxings have a higher degree of influence on MASs, and the coefficient of the number of martial arts boxings is as high as 0.542, which is the most important influencing factor. The per capita disposable income of each province has the least negatively influence on the number of MASs, which may be caused by the differences in the cognition and acceptance of Wushu among people in different regions (Table 4). Additionally, the indicators related to the hometowns of martial arts were impacted during the factor analysis. As a result, they were visualized and analyzed in the form of images (Fig 6).

Based on the global parameters obtained by OLS model regression (Table 5), the geographical weighted regression model (GWR4.0.90) was further used to analyze the spatial distribution heterogeneity and local characteristics. The bandwidth, geographical range and local regression parameters were obtained by GWR calculation on the provincial Wushu intangible cultural heritage of boxings, martial arts boxings, and population education level (Table 6).

The $R^2$ values (Table 6) obtained by GWR model and the revised $R^2$ values are 0.801 and 0.752 respectively, which are greatly improved compared with the OLS global regression parameters (Table 5), and the fitting degree is better. In OLS global regression parameters, the Classic AIC value and AICc value are 51.892 and 60.437, respectively. After GWR, the Classic AIC value and AICc value are 53.134 and 60.291, respectively. The likelihood of $-2$ log-likelihood in GWR model is $-18.245$, compared with the likelihood of $-2$ log-likelihood in OLS model is $-18.946$. To sum up, GWR model is suitable for this study than OLS model (Table 6).

## 5. Discussion

### 5.1. The association between MASs and Wushu hometowns

This paper utilizes ArcGIS software to provide a visual representation of the spatial distribution patterns of 88 Wushu hometowns and 492 MASs across mainland China (Fig 6). While the Wushu hometowns are depicted as planar entities, the MASs are pinpointed, revealing a clear spatial alignment between the two. The results confirm the objectivity of the data sources used in the analysis, as both the Wushu hometowns and MASs exhibit a strong correlation in their geographical distribution at the provincial level [24]. This alignment underscores the close relationship between the development of martial arts schools and the cultural heritage of Wushu hometowns [24,38]. Unlike previous studies that primarily focused on analyzing the factors influencing the development of martial arts schools [39,40], this paper introduces a visual analysis of how Wushu hometowns provide cultural and spatial support to MASs. The spatial alignment between Wushu

**Table 4.  The results of OLS model.**

| Variate | Coefficient | Standard deviation | T value | Robust P-value | VIF |
|---|---|---|---|---|---|
| Constant Term | 6.017011 | 5.392201 | 1.115873 | 0.134970 | |
| Provincial Wushu intangible cultural heritage of boxings | 0.509454 | 0.190033 | 2.680870 | 0.01860* | 6.041 |
| Population Education Level (College degree or above) | 0.020295 | 0.009406 | 0.932991 | 0.242042* | 9. 291 |
| Population density | 0.000000 | 0.000000 | 0.008877 | 0.990592 | 10.385 |
| Martial arts boxings | 0.541693 | 0.183545 | 2.951238 | 0.003497* | 4.494 |
| Per capita disposable income of each province | −0.000334 | 0.000177 | −1.892011 | 0.016503* | 2.181 |
| $R^2$ | 0.852620 | | | | |
| Correction $R^2$ | 0.815775 | | | | |
| Joint chi-square statistics | 23.140654 | | | | |
| Koenker(BP) statistics | 6.876685 | | | | |

When the Koender (BP) statistic is significant, it is necessary to judge the statistical significance of the independent variable by robust P-value; *, **, and *** indicate that the variable is statistically significant at the 10%, 5%, and 1% levels, respectively.

hometowns and MASs highlights the close relationship between the development and sustainability of MASs and the cultural heritage of these regions. Wushu hometowns play a crucial role in nurturing martial arts talent, promoting Wushu culture, and preserving the traditions of martial arts. The historical and cultural significance of these hometowns forms the foundation of their vital support in advancing martial arts education. These regions not only offer a rich pool of skilled instructors but also attract high-quality students to MASs, further strengthening the connection between the spatial distribution of Wushu hometowns and the flourishing growth of MASs. The findings of this research not only reveal a strong correlation between MASs and Wushu hometowns but also provide valuable insights for optimizing the spatial layout of MASs. Based on these results, the study offers the following recommendations:

(1) Governments at various levels should consider establishing special martial arts education zones in regions surrounding Wushu hometowns. These zones should receive tailored policy support, including preferential funding, to foster the growth of MASs in areas with rich cultural heritage.

(2) MASs should be encouraged to establish close collaborative relationships with Wushu hometowns, forming a "school + base" model for coordinated development. This model leverages the cultural resources and talent pool of Wushu hometowns to enhance both the educational quality and cultural depth of MASs.

(3) MASs should harness the unique cultural characteristics of Wushu hometowns to develop distinct educational brands. For instance, MASs near Shaolin Temple can highlight the fusion of 'Zen' and martial arts, while those near the birthplace of 'Tai Chi' can emphasize the philosophy of martial arts as a form of wellness.

(4) Establishment of Martial Arts Talent Training Bases: Martial arts talent training bases should be set up in Wushu hometowns to supply high-quality instructors and student recruitment channels for surrounding MASs. This initiative would ensure a steady flow of skilled professionals and dedicated students to support the schools' growth.

## 5.2.  The correlation between MASs and the educational level of population

In China's MASs and halls, 58% of enrolled students come from agricultural backgrounds, and the educational qualifications of MASs instructors typically do not exceed the undergraduate

# The Distribution of Martial Arts Schools and Wushu Hometowns in China

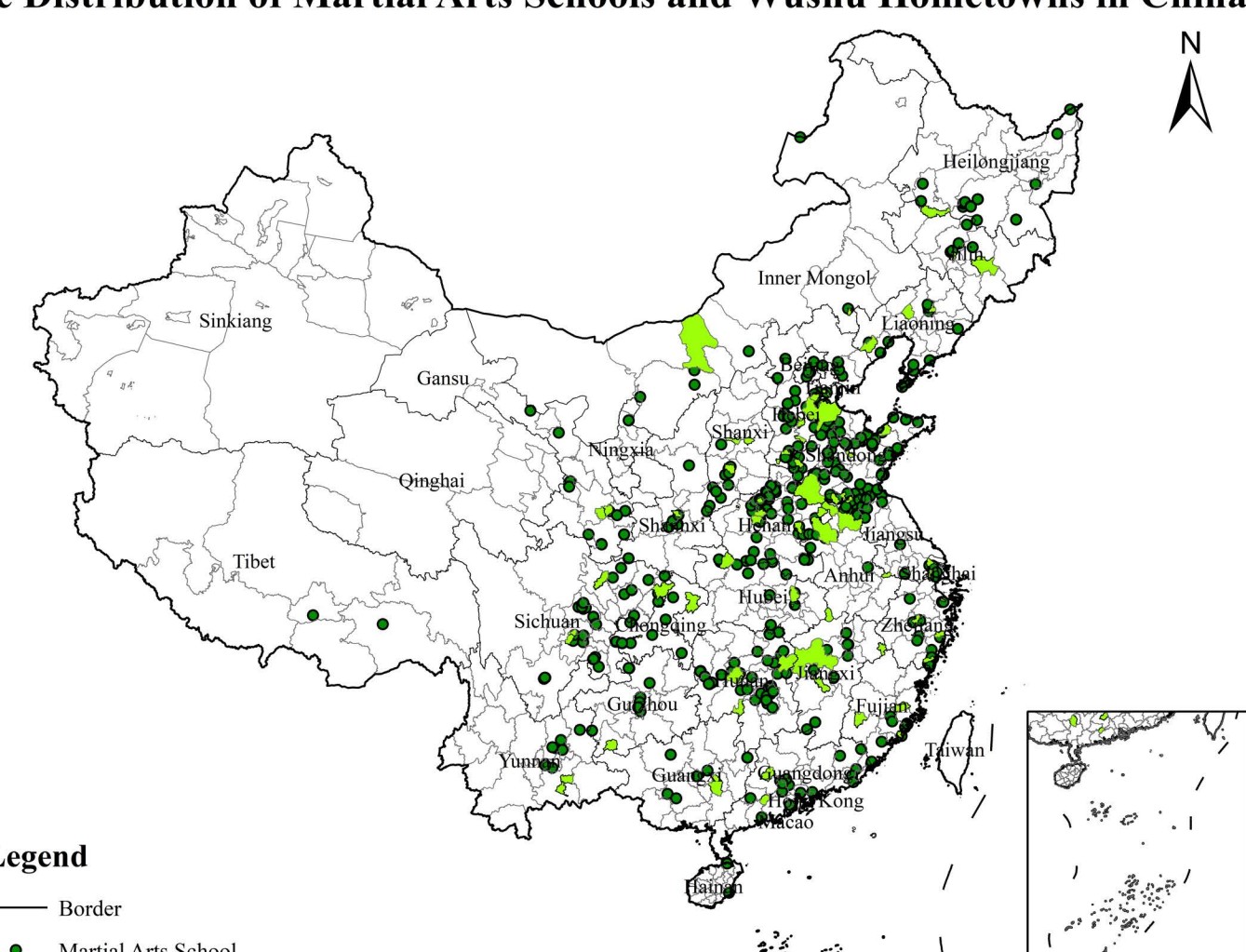

**Fig 6. The distribution of martial arts schools and Wushu hometowns in China** (http://bzdt.ch.mnr.gov.cn/browse.html?picId=%224o28b0625501a-d13015501ad2bfc0272%22). (A) The map is based on the standard map service website of the National Bureau of Surveying, Mapping and Geographic Information Service No.GS (2019)1682 standard map drawing, the base map has not been modified. (B) The map projection for the martial arts school was obtained from the Baidu Map API using latitude and longitude data in Python. (C) The map projection for the Wushu hometowns was obtained from the 88 Wushu hometowns data in China. (D) Excluding data on Wushu hometowns in Hong Kong, Macau, and Taiwan.

**Table 5. Global regression parameters of OLS model.**

| Number | Parameter | Value |
|---|---|---|
| 1 | Residual sum of squares | 6.6162 |
| 2 | −2 log-likelihood | −18.946 |
| 3 | Classic AIC | 51.892 |
| 4 | AICc | 60.437 |
| 5 | R square | 0.801 |
| 6 | Adjusted R square | 0.752 |

Table 6. Regional regression parameters of GWR model.

| Number | Parameter | Value |
|---|---|---|
| 1 | Residual sum of squares | 5.890 |
| 2 | Effective number of parameters (trace(s)) | 7.322 |
| 3 | Degree of freedom (n-trace(s)) | 23.678 |
| 4 | ML based sigma estimate | 0.499 |
| 5 | −2 log-likelihood | −18.245 |
| 6 | Degree of Dependency (DoD) | 0.987 |
| 7 | Classic AIC | 53.134 |
| 8 | AICc | 60.291 |
| 9 | BIC/MDL | 65.068 |
| 10 | R square | 0.810 |
| 11 | Adjusted R square | 0.749 |

level, with minimal involvement in higher education [38]. The OLS model reveals a weakly positive correlation (correlation coefficient: 0.020295) between the population's educational attainment and the prevalence of MASs, suggesting that martial arts institutions are less prominent in regions with higher educational levels [13,41]. This weak correlation between MASs and higher education may stem from increasing educational qualifications, rising vocational expectations, and growing demand for specialized skills. Contemporary MASs primarily target individuals outside the mainstream primary, secondary, and tertiary educational systems, with limited engagement from those pursuing higher education. This aligns with previous analyses of employment outcomes for MASs graduates [40,42,43]. As educational levels increase, engagement with martial arts education diminishes, particularly within higher education, where martial arts are typically integrated into broader programs such as ethnic traditional sports, physical education, and sports training, rather than offered through dedicated martial arts institutions. Furthermore, the competitive nature of higher education admissions results in few MASs graduates securing advanced academic opportunities. This trend is evident in the selection processes for advanced degrees in national traditional sports, as corroborated by the OLS model findings. As human capital distribution improves and educational standards rise, the qualifications and employment prospects within martial arts education are increasingly emphasized. Looking ahead, the quality of instruction in MASs may become a critical constraint on their development. To address this, MASs should consider developing a diversified curriculum system to cater to students from different educational backgrounds. This can be achieved by collaborating with higher education institutions to create interdisciplinary programs that combine martial arts with modern disciplines such as sports science, cultural studies, and tourism management, thereby enhancing the academic and contemporary relevance of martial arts education. Additionally, the delayed recognition of traditional national sports science as an academic discipline, along with misalignments in disciplinary focus and educational paradigms, may further exacerbate these challenges [14,44].

## 5.3. The correlation between MASs and population density

The results derived from the OLS model indicate that the spatial distribution of Martial Arts Schools (MASs) shows no statistically significant correlation with population density, as the P-value is insignificant, and the correlation coefficient is effectively zero. This outcome may stem from limitations in the chosen indicator system. Nevertheless, it is evident from statistical data that population distribution in China is highly imbalanced (https://www.gov.cn/guoqing/2021-05/13/content_5606149.htm).

This uneven population distribution significantly affects the allocation of educational resources, the structure of educational systems, and the distribution of educational facilities, which in turn influences the spatial distribution of MASs. Provinces like Henan and Shandong, with high population densities, exemplify regions where the substantial demographic base intensifies competition for resources and increases educational demand, leading to knowledge spillover effects. In such densely populated areas, MASs flourish due to an ample supply of students, abundant teaching staff, and strong financial support. In contrast, in sparsely populated provinces such as Xinjiang, Qinghai, Tibet, and Inner Mongolia (Fig 4 Hu's line), the lack of sufficient student populations and foundational educational infrastructure hinders the development and spread of MASs, confining them to slower developmental progress. This interpretation underscores the potential impact of population distribution on the uneven allocation of educational resources, which, although not captured statistically by the OLS model, remains an important factor influencing the spatial patterns of martial arts schools across different provinces. Therefore, this study proposes the following recommendations:

(1) MASs should pursue large-scale and diversified development in densely populated areas, while focusing on specialized, high-quality development in sparsely populated regions.

(2) In densely populated areas, an educational network should be established, consisting of core schools and branch institutions to create a ripple effect. In sparsely populated regions, a "central school + satellite teaching points" model should be adopted to enhance the efficiency of educational resource utilization.

(3) Modern information technology should be leveraged to develop online martial arts education platforms, breaking geographical barriers and providing high-quality martial arts education resources to sparsely populated areas.5.4. The correlation between MASs and martial arts boxings.

Since provincial Wushu intangible cultural heritage boxings and Wushu boxings were the same influencing factors, the analysis was combined. Findings from the OLS regression model reveals a robust correlation coefficient of 0.541693 and 0.509454 between the array of martial arts disciplines, known as 'martial arts boxings', and the spatial distribution of MASs, signifying a substantial interdependence. The profusion and diversity of martial arts disciplines are integral to the fabric and evolution of these schools. Dually, this rich tapestry of martial arts styles addresses the multifaceted educational desires of students and underscores the formidable prowess and unique identity of each school. Take, for instance, the prestigious Shaolin Wushu school in Henan, which offers a venerable array of disciplines such as Shaolin boxing, Taizu boxing, ground trip boxing, drunken boxing, and Arhat boxing [3,45]. These boxings of Wushu have a magnetic appeal, drawing a multitude of students from both domestic and international locales, thus serving as a testament to the school's robust instructional cadre and accentuating its distinctive educational offerings. The dynamic interplay of these factors indisputably creates a conducive environment for the propagation and endurance of martial arts disciplines and MASs alike. Yet, it is imperative to acknowledge that an augmentation in the diversity of martial arts boxings necessitates substantial investments in resources, inflates operational costs, and intensifies the demand for skilled instructors, potentially leading to resource redundancies and disparities in instructional quality. Consequently, this poses multifaceted challenges. Overall, the multiplicity of martial arts disciplines exerts a nuanced impact on the operations of MASs [13,46]. Based on results, the study offers the following recommendations:

(1) A distinctive educational brand should be created at the characteristic of regional Wushu boxings.

(2) A curriculum system comprising "core Wushu styles and multiple elective courses" should be developed, not only to preserve traditional characteristics but also to cater to the

diverse needs of students. It is essential to protect and preserve rare Wushu styles, while also innovating on the foundation of tradition to develop new martial arts programs that align with the demands of modern society.

(3) It is recommended that a "Martial Arts Education Alliance" be established in regions rich in Wushu traditions to promote communication and integration among different martial arts styles. This would help elevate the overall standard of martial arts education, capitalize on the unique qualities of various Wushu styles, formulate targeted promotion strategies, and enhance the influence and recognition of martial arts.

### 5.5. The correlation between MASs and provincial GDP

Based on the results of the OLS model and the regional clustering characteristics of MASs, there is no significant negative correlation between the spatial distribution of these schools and the level of regional economic development. The OLS model's computation yields a correlation coefficient of −0.000334, with a P-value indicating statistical significance at the 10% level. The implementation of the "Nine-Year Compulsory Education" policy, and the "Two Free and One Subsidy" initiative (http://www.moe.gov.cn//jyb_xwfb/xw_zt/moe_357/s3580/moe_2448/moe_2450/moe_2459/tnull_33420.html), has placed MASs at a disadvantage within the highly competitive educational landscape. Due to high tuition fees and relatively low academic standards, these schools face difficulties in attracting high-quality students. Considering the relevant educational policies introduced in China over the past five years, the relationship between the spatial distribution of MASs and provincial GDP remains weak. The findings suggest that factors beyond economic development—such as educational policies, cultural preservation efforts, and demographic characteristics—play a more critical role in influencing the geographical distribution and sustainability of martial arts schools across different provinces. This study proposes the following recommendations:

(1) It is recommended that the government prioritize cultural heritage preservation and social benefits when formulating martial arts education policies, rather than focusing solely on economic gains.

(2) Diverse financing channels, such as public donations and corporate sponsorships, should be explored to reduce reliance on the local economic development level.

(3) Efforts should be made to actively develop ancillary industries to diversify income sources and enhance the sustainability of martial arts education.

(4) The study calls for both national and local governments to incorporate martial arts education into cultural industry development plans, providing financial and policy support to promote the balanced development of martial arts education.

(5) MASs in economically developed regions should be encouraged to collaborate with those in less developed areas. Through partnerships, resource sharing, and support programs, the coordinated development of martial arts education can be promoted.

## 6. Conclusion

The main conclusions of this study are as follows:

(1) The national MASs, as education institutions imbued with the essence of traditional culture, philosophy, and psychology, offer a curriculum rich in heritage yet complex in its educational nature. This intricacy inherently amplifies the challenges faced in administration.

It becomes imperative, therefore, to speed up standardizing the establishment of MASs, enhancing their academic stature, and steering their developmental trajectories. Such measures are essential not only to bolster the institutional framework but also to provide robust support and technical resources crucial for nurturing talent within the realms of martial arts and the disciplines of ethnic traditional sports. MASs belong to the spatial agglomeration distribution type, showing unbalanced distribution characteristics. The main concentration in Shandong, Henan, Hebei, Hunan, and Sichuan with sparse distribution in the western and northern regions and discrete distribution in the southeast. This distribution characteristic indicates that the number and scale of Chinese MASs vary greatly among different regions, and the fundamental reason for this situation is the unbalanced distribution of martial arts cultural resources. Therefore, in the layout and development of MASs, it is necessary to fully consider the compound influence of multiple factors such as geographical location, economic development, and cultural history, to realize the reasonable distribution of MASs based on making full use of regional martial arts cultural resources.

(2) The distribution of MASs in China has one high-density region and two sub-high-density regions, with Shandong as the center connecting Henan and Hebei to form a high-density region of MASs distribution; The two sub-high density regions, with Sichuan and Hunan as the center respectively, show an overall distribution trend of radiation spreading from the north and south regions to the surrounding areas, which may reflect the influence of population development level and economic and social pattern. These findings indicate that population, economic and social factors should be considered in the future layout and development strategy of MASs to achieve a more optimized and balanced spatial distribution of MASs.

(3) This study deeply discusses the multiple factors affecting the spatial distribution of MASs, especially the number of martial arts boxings, the education level of the population in the Wushu hometowns, population density and other factors, providing certain theoretical support and reference for the regional layout of MASs. In addition, due to the unique regional characteristics of China and the influence of semi-closed rural society, martial arts culture has obvious regional characteristics. For example, Laizhou MAS in Shandong Province, Heze MAS, Shaolin Temple in Henan Province, Tagou MAS in Henan Province, Hualong MAS in Hebei Province, and other MASs are the characteristics of long-term accumulation of regional culture. However, other factors that may affect the spatial distribution of MASs, such as socio-economic conditions, educational input, and policy environment, are not deeply discussed in this study, which may limit the depth and breadth of a comprehensive understanding of the spatial distribution of MASs. Future studies can further consider and analyze these factors to understand and explain the spatial distribution of MASs more fully. For example, considering the economic development of the location of a MAS may further explain the geographical distribution of MASs. If a region has better economic conditions, it may have more resources to invest in the construction and development of MASs, which may lead to a higher number of it in the region. In addition, the policy environment may also affect the spatial distribution of MASs. If the government gives more support and encouragement in policy, then there may be more MASs in these places. In general, the study on the spatial distribution of MASs should consider the influencing factors more comprehensively, and not be limited to the number of martial arts, Wushu hometowns, population education level, intangible cultural heritage boxings of Wushu, population education level, Per capita disposable income, and population density. This will help us to better understand and explain the spatial distribution of MASs and provide more scientific guidance for the development and layout of MASs.

## 7. Summary

In the grand schema of cultivating distinguished traditional cultural disciplines and forging a high-caliber educational system, the sector of martial arts education warrants substantial attention. MASs ought to be recognized as pivotal hubs for the preservation and dissemination of CMAs heritage. Under the background of fierce cultural competition, it is incumbent upon us to escalate investments in the educational infrastructure of these schools, to innovate their curricula, seamlessly blending martial arts with the quintessence of traditional culture, and to craft a core curriculum that synergizes technique with cultural education. This approach aims to transcend the antiquated notion that MASs are merely sanctuaries for the underprivileged. Rather, they stand at the vanguard of safeguarding and advancing the rich tapestry of traditional Chinese culture. To this end, it is essential for policymakers to promulgate reforms targeting the strategic orientation of MASs, to normalize their standards, to revitalize the teaching milieu, to enhance the welfare of educators, and to underscore the foundational significance of martial arts institutions in cultivating adept personnel, fostering martial arts culture, and spurring the growth of the martial arts industry. These measures will collectively consolidate the essential function of MASs as bastions of cultural excellence and pedagogical innovation.

## Author contributions

**Conceptualization:** Pengfei Yu.

**Data curation:** Pengfei Yu.

**Formal analysis:** Jianliang Guan.

**Investigation:** Pengfei Yu.

**Methodology:** Jianliang Guan, Guohua Chen.

**Project administration:** Pengfei Yu.

**Software:** Jianliang Guan.

**Validation:** Qi Guo.

**Writing – original draft:** Pengfei Yu.

**Writing – review & editing:** Xiaoming Yang, Qi Guo, Jianliang Guan.

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
