## [Decision Letter · Decision Letter 0]

23 Aug 2024

PONE-D-24-30655Spatial Distribution Characteristics and Influencing Factors of China Martial Arts Schools Based on Baidu Map APIPLOS ONE

Dear Dr. Yu,

Thank you for submitting your manuscript to PLOS ONE. After careful consideration, we feel that it has merit but does not fully meet PLOS ONE’s publication criteria as it currently stands. Therefore, we invite you to submit a revised version of the manuscript that addresses the points raised during the review process.

We look forward to receiving your revised manuscript.

Kind regards,

Yaodong Gu

Academic Editor

PLOS ONE

Journal Requirements:

2. We note that your Data Availability Statement is currently as follows: All relevant data are within the manuscript and its Supporting Information files. the Baidu Map API data URL is from (https://api.map.baidu.com/lbsapi/getpoint/)

4. Please ensure that you include a title page within your main document. You should list all authors and all affiliations as per our author instructions and clearly indicate the corresponding author.

5. We note that Figures 2,4,5 and 6 in your submission contain map/satellite images which may be copyrighted. All PLOS content is published under the Creative Commons Attribution License (CC BY 4.0), which means that the manuscript, images, and Supporting Information files will be freely available online, and any third party is permitted to access, download, copy, distribute, and use these materials in any way, even commercially, with proper attribution. For these reasons, we cannot publish previously copyrighted maps or satellite images created using proprietary data, such as Google software (Google Maps, Street View, and Earth). For more information, see our copyright guidelines: http://journals.plos.org/plosone/s/licenses-and-copyright.

a. You may seek permission from the original copyright holder of Figures 2,4,5 and 6 to publish the content specifically under the CC BY 4.0 license.  

6. Please remove your figures from within your manuscript file, leaving only the individual TIFF/EPS image files, uploaded separately. These will be automatically included in the reviewers’ PDF.

Additional Editor Comments:

The disucssion part shall be more deeply discussed.

Reviewers' comments:

Reviewer's Responses to Questions

**Comments to the Author**

1. Is the manuscript technically sound, and do the data support the conclusions?

Reviewer #1: Yes

Reviewer #2: Yes

2. Has the statistical analysis been performed appropriately and rigorously? 

Reviewer #1: Yes

Reviewer #2: Yes

3. Have the authors made all data underlying the findings in their manuscript fully available?

Reviewer #1: Yes

Reviewer #2: No

4. Is the manuscript presented in an intelligible fashion and written in standard English?

Reviewer #1: Yes

Reviewer #2: No

5. Review Comments to the Author

Reviewer #1: This paper is novel as a whole. The research data and results are relatively new. The existing problem is that the research is not deep enough.

The problems in this paper are as follows:

1. “the spatial distribution of the herbaceous marsh vegetation, ecological effects of land-use change on two sides of the Hu’s Line”, these two research documents are not closely related to the research content of this paper, and it is suggested to delete them.

2. For Introduction section, the summary and analysis of the relevant research status at home and abroad are seriously inadequate. The authors should vigorously quote and expound relevant research documents, and summarize and concise the technical problems existing in these research documents.

3. For "Geological Information", it should be Geographical, not Geological.

4. For "Feb 2024", this month is inconsistent with the month in the abstract and should be unified and consistent.

5. For the sections from 2.2.1 to 2.2.6, each section should give the number of the cited references.

6. The Table 1 should give the number of the cited references.

7. For the Figure 1, the graphic meaning of the figure is not corresponding to the content of the paper, and it is not clear, so the authors should briefly explain the figure.

8. In the section 4.1, among the influencing factors, the authors did not consider the economic factor as its obvious deficiency.

9. In the section 4.2, the number of influencing factors is small. The authors should choose more influencing factors, such as economy and culture.

10. In the section 5, the research results mentioned above in this paper should be compared and discussed with the research results of related research literature in terms of correctness and accuracy.

11. In the section 6, for "socio-economic conditions", this is a very important influencing factor, and it is suggested that it should be added to the study of influencing factors in this paper.

Reviewer #2: 1.Although the paper provides some research background, it is better to more comprehensively review existing studies on the spatial distribution of martial arts schools and their influencing factors.

2.Please clarify how the sample of martial arts schools was selected and whether it is representative. Additionally, discuss the potential impact of regions not included in the study on the overall conclusions.

3.The discussion section should more comprehensively explore the implications, limitations, and future research directions of the study. For example, consider discussing how the findings could be applied to the optimization of martial arts school layout and policy formulation.

4.In the geographic weighted regression analysis section, could you explain the rationale for choosing these specific variables as independent variables?

6. PLOS authors have the option to publish the peer review history of their article (what does this mean? ). If published, this will include your full peer review and any attached files.

**Do you want your identity to be public for this peer review?** For information about this choice, including consent withdrawal, please see our Privacy Policy .

Reviewer #1: No

Reviewer #2: No

---

## [Author Response · Author response to Decision Letter 0]

20 Sep 2024

I have uploaded the picture file as required

---

## [Decision Letter · Decision Letter 1]

14 Oct 2024

PONE-D-24-30655R1Spatial Distribution Characteristics and Influencing Factors of China Martial Arts Schools Based on Baidu Map APIPLOS ONE

Dear Dr. Yu,

Thank you for submitting your manuscript to PLOS ONE. After careful consideration, we feel that it has merit but does not fully meet PLOS ONE’s publication criteria as it currently stands. Therefore, we invite you to submit a revised version of the manuscript that addresses the points raised during the review process.

We look forward to receiving your revised manuscript.

Kind regards,

Yaodong Gu

Academic Editor

PLOS ONE

**Journal Requirements:**

**Additional Editor Comments:**

Please check some minor questions raised by the reviewer.

Reviewers' comments:

Reviewer's Responses to Questions

**Comments to the Author**

1. If the authors have adequately addressed your comments raised in a previous round of review and you feel that this manuscript is now acceptable for publication, you may indicate that here to bypass the “Comments to the Author” section, enter your conflict of interest statement in the “Confidential to Editor” section, and submit your "Accept" recommendation.

Reviewer #2: All comments have been addressed

Reviewer #3: (No Response)

2. Is the manuscript technically sound, and do the data support the conclusions?

Reviewer #2: (No Response)

Reviewer #3: Yes

3. Has the statistical analysis been performed appropriately and rigorously? 

Reviewer #2: (No Response)

Reviewer #3: Yes

4. Have the authors made all data underlying the findings in their manuscript fully available?

Reviewer #2: (No Response)

Reviewer #3: Yes

5. Is the manuscript presented in an intelligible fashion and written in standard English?

Reviewer #2: (No Response)

Reviewer #3: Yes

6. Review Comments to the Author

**Reviewer #2: ** (No Response)

**Reviewer #3: ** Review comment

this paper presents a comprehensive investigation into the spatial distribution characteristics and influencing factors of Martial Arts Schools (MASs) in China using Baidu Map API and a geographically weighted regression model, which adds a valuable and innovative contribution to the field. The authors effectively utilized Python programming to collect a large dataset of geographical information, and the use of ArcGIS for analysis provided a thorough quantitative and qualitative assessment. The findings highlight the clear regional clustering of MASs, particularly in provinces like Shandong and Henan, which are deeply connected to China’s martial arts cultural heritage, history, population, and economic development. However, several areas could be improved:

Specific comments

1. “The spatial analytical endeavor unveiled a Moran's I value of 0.172, accompanied by a Z-score of 1.75 and a P-value of 0.079, signifying an uneven and clustered distribution pattern predominantly concentrated in provinces such as Shandong, Henan, Hebei, Hunan, and Sichuan.” Could you explain why a Moran's I value of 0.172 was considered significant, given that the P-value is above the conventional threshold of 0.05?

2. In the Introduction part, “Martial Arts Schools (MASs) serve as crucial institutions for the preservation and dissemination of wushu culture, acting as the primary platform for the development of basic martial arts education in China.” The introduction mentions the marginalization of martial arts within China's compulsory education system. Could you expand on how this marginalization has evolved over the years.

3. Abbreviations that exist in articles should be explained when thy first appear, such as Martial Arts Schools (MASs). Please give official explanations of the abbreviations in the article.

4. In the Data and Method part, “We chose ‘martial arts school’ and ‘wushu school’ as the search terms; Leveraging the Python programming language, an extensive dataset encompassing the geographical metadata of 1,986 MASs was meticulously harvested from Baidu Maps.” How did the authors validate the accuracy of the geographical data collected via Baidu Maps API? Were there any limitations identified in the dataset that could affect the analysis?

5. In “the Spatial Distribution Results Analysis of MASs” part, “The spatial distribution analysis of (MASs) reveals a Z-score of 1.75390, indicating that the probability of this spatial distribution occurring randomly is less than 10%.” How do the authors explain the geographical concentration of MASs in provinces like Shandong and Henan, and does this concentration align with historical trends in martial arts education?

6. In “The Selection of Influencing Factors and the Analysis of Geographical Weighted Regression Results” part. “The number of martial arts boxings, and the number of provincial wushu intangible cultural heritage of boxings have a higher degree of influence on MASs, and the coefficient of the number of martial arts boxings is as high as 0.542, which is the most important influencing factor.” How did the authors ensure that multicollinearity between variables did not affect the regression results, particularly between cultural heritage and martial arts boxings?

7. PLOS authors have the option to publish the peer review history of their article (what does this mean? ). If published, this will include your full peer review and any attached files.

**Do you want your identity to be public for this peer review?** For information about this choice, including consent withdrawal, please see our Privacy Policy .

Reviewer #2: No

Reviewer #3: **Yes: ** Zixiang Gao

---

## [Decision Letter · Decision Letter 2]

13 Nov 2024

Spatial Distribution Characteristics and Influencing Factors of China Martial Arts Schools Based on Baidu Map API

PONE-D-24-30655R2

Dear Dr. Yu,

We’re pleased to inform you that your manuscript has been judged scientifically suitable for publication and will be formally accepted for publication once it meets all outstanding technical requirements.

Kind regards,

Yaodong Gu

Academic Editor

PLOS ONE

Additional Editor Comments (optional):

Well done.

Reviewers' comments:

Reviewer's Responses to Questions

**Comments to the Author**

1. If the authors have adequately addressed your comments raised in a previous round of review and you feel that this manuscript is now acceptable for publication, you may indicate that here to bypass the “Comments to the Author” section, enter your conflict of interest statement in the “Confidential to Editor” section, and submit your "Accept" recommendation.

Reviewer #3: (No Response)

2. Is the manuscript technically sound, and do the data support the conclusions?

Reviewer #3: (No Response)

3. Has the statistical analysis been performed appropriately and rigorously? 

Reviewer #3: (No Response)

4. Have the authors made all data underlying the findings in their manuscript fully available?

Reviewer #3: (No Response)

5. Is the manuscript presented in an intelligible fashion and written in standard English?

Reviewer #3: (No Response)

6. Review Comments to the Author

Reviewer #3: (No Response)

7. PLOS authors have the option to publish the peer review history of their article (what does this mean? ). If published, this will include your full peer review and any attached files.

**Do you want your identity to be public for this peer review?** For information about this choice, including consent withdrawal, please see our Privacy Policy .

Reviewer #3: **Yes: ** Zixiang Gao

---

## [Editor Report · Acceptance letter]

PONE-D-24-30655R2

PLOS ONE

Dear Dr. Yang,

I'm pleased to inform you that your manuscript has been deemed suitable for publication in PLOS ONE. Congratulations! Your manuscript is now being handed over to our production team.

Kind regards,

on behalf of

Professor Yaodong Gu

Academic Editor

PLOS ONE